# Modeling bee movement shows how a perceptual masking effect can influence flower discovery

**Ana Morán**[ID]*, **Mathieu Lihoreau, Alfonso Pérez-Escudero**\*[◉], **Jacques Gautrais**[ID]\*[◉]

Centre de Recherches sur la Cognition Animale (CRCA), Centre de Biologie Intégrative (CBI), Université de Toulouse, CNRS, UPS, 118 route de Narbonne, Toulouse, France

◉ These authors contributed equally to this work.

\* ana.moran-hernandez@univ-tlse3.fr (AM); alfonso.perez-escudero@univ-tlse3.fr (APE); jacques.gautrais@univ-tlse3.fr (JG)

**Data Availability Statement:** All relevant data are within the paper and its Supporting Information files.

## Abstract

Understanding how pollinators move across space is key to understanding plant mating patterns. Bees are typically assumed to search for flowers randomly or using simple movement rules, so that the probability of discovering a flower should primarily depend on its distance to the nest. However, experimental work shows this is not always the case. Here, we explored the influence of flower size and density on their probability of being discovered by bees by developing a movement model of central place foraging bees, based on experimental data collected on bumblebees. Our model produces realistic bee trajectories by taking into account the autocorrelation of the bee's angular speed, the attraction to the nest (homing), and a gaussian noise. Simulations revealed a « masking effect » that reduces the detection of flowers close to another, with potential far reaching consequences on plant-pollinator interactions. At the plant level, flowers distant to the nest were more often discovered by bees in low density environments. At the bee colony level, foragers found more flowers when they were small and at medium densities. Our results indicate that the processes of search and discovery of resources are potentially more complex than usually assumed, and question the importance of resource distribution and abundance on bee foraging success and plant pollination.

## Author summary

Understanding how pollinators move in space is key to understand plant reproduction and its consequences on terrestrial ecosystems. Current models assume simple movement rules that predict flowers are more likely to be visited—and hence pollinated—the closer they are to the pollinators' nest. Here we developed an explicit movement model that incorporates realistic features of bumblebee behaviour, and calibrated it with experimental data collected in naturalistic conditions. Our model shows that the probability to visit a flower does not only depend on its position, but also on the position of other flowers around that may mask it from the forager. This perceptual masking effect means that

**Funding:** AM was supported by a PhD Fellowship from the French Government. ML was supported by grants of the Agence Nationale de la Recherche (3DNaviBee ANR-19-CE37-0024), and the European Commission (FEDER ECONECT MP0021763, ERC Cog BEE-MOVE GA101002644). APE acknowledges funding from a CNRS Momentum grant (https://www.cnrs.fr/) and a Fyssen Foundation Research grant (https://www.fondationfyssen.fr/en/). The funders had no role in study design, data collection and analysis, decision to publish, or preparation of the manuscript.

**Competing interests:** The authors have declared that no competing interests exist.

pollination efficiency depends on the density and spatial arrangement of flowers around the pollinators' nest, often in counter-intuitive ways. Taking these effects into account may be key for improving practical actions in precision pollination and pollinator conservation.

## Introduction

Pollinators, such as bees, wasps, flies, butterflies, but also bats and birds, mediate a key ecosystemic service on which most terrestrial plants and animals, including us humans, rely on. When foraging for nectar, animals transfer pollen between flowers, which mediates plant reproduction. Understanding how pollinators move, find and choose flowers is thus a key challenge of pollination ecology [1]. In particular, this may help predict and act on complex pollination processes in a context of a looming crisis, when food demand increases and populations of pollinators decline [2,3].

Foraging pollinators have long been assumed to move randomly [4–7] or use hard wired movement rules such as visiting the nearest unvisited flower [8], exploiting flower patches in straight line movements [9], navigating inflorescences from bottom to top flowers [10], or using win-stay lose-leave strategies [11]. Accordingly pollination models relying on these observations typically predict diffusive movements in every direction [12]. However, recent behavioural research shows this is not true when animals forage across large spatial scales [13]. In particular, studies using radars to monitor the long distance flight paths of bees foraging in the field demonstrate that foragers learn features of their environment to navigate across landscapes and to return to known feeding locations [14,15]. This enables them to develop shortcuts between feeding sites [16] and use efficient multi-destination routes (traplines) minimizing overall travel distances [17,18]. These routes are re-adjusted each time a feeding site is depleted and new ones are discovered [19].

How bees learn such foraging routes has been modelled using algorithms implementing spatial learning and memory [20–22]. While this has greatly advanced our understanding of bee exploitative movements patterns, none of these models have looked at search behaviour, either assuming insects already know the locations of all available feeding sites in their environment or discover them according to fixed probabilistic laws (e.g. the probability to discover a flower at a given location is proportional to $1/L^2$ where $L$ represents the distance to that flower [20–22]).

However, an increasing number of experimental data indicates that this is not the case. Firstly, bees, like many pollinators, are central place foragers so that every foraging trip starts and ends at the nest site [23]. This implies that their range of action is limited. Recordings of bee search flights show how individuals tend to make loops centered at the nest when exploring a new environment and look for flowers [14,23]. These looping movements are not compatible with the assumption that bees make diffusive random walks or Lévy flights [24,25]. Secondly, the spatial structure of the foraging environment itself may also greatly influence flower discovery by bees. In particular, the probability of finding a flower heavily depends on the location of the flower visited just before, ultimately influencing the direction and geometry of the routes developed by individuals [17–20,26,27]. Since bees are more attracted by large flowers than by smaller ones [28], this suggests that small isolated flowers could be missed if they are located next to a larger patch. Such « masking effect » on the probability to visit specific flowers depending on the presence of other flowers around could have important consequences for bee foraging success, for instance by precluding the discovery of some highly

rewarding flowers. This could also influence plant pollination, if bees are spatially constrained to single flower patches and plant outcrossing is limited.

Here we explored these potential effects by developing a model of bee search movement simulating the tendency of bumblebees to make loops around their nest. We used our model to examine the probability for bees to discover flowers in environments defined by resources of various sizes and abundances. We hypothesized that looping movements characteristic of bee exploratory flights combined with perceptual masking effects by which the probability of finding given flowers is affected by the presence of others, would result in strikingly different predictions for flower discovery rates than the typical diffusive random walk movements.

## Model background

Our model was designed to describe exploratory behaviour of bees, render realistic trajectories with regard to their motion and account for their "central place foraging" constraints to remain in the vicinity of their nest, with regular returns to it. We used bumblebees as model system to calibrate our bee movement model since their flights have been best described in the field [14,17–20,26–29] and we had access to experimental data (see below). We then needed to compute the probability of finding a given flower, a problem that pertains to the field of First Passage Time statistics and splitting probabilities (i.e. whether a target is hit sooner than another one) realized by a random walker with range-limited trajectories [30]. Two main approaches exist for rendering exploratory trajectories of animals (or any active matter): the Active Brownian Particle [31] and the Persistent Random Walker (or run-and-tumble [32]).

## Active Brownian Motion

Active Brownian Motion and its derivatives describe trajectories by a stochastic process governing the particle's velocity [33]. These models can incorporate a central place foraging component through two main mechanisms: harmonic potentials and stochastic resetting.

Harmonic potentials trap the particle in the vicinity of the trap center, yielding a Non Equilibrium Stationary State (NESS) in the large-time limit [34–40] with particles orbiting around the center [33]. Stochastic resetting forces the particle back to the origin periodically [41], either instantaneously (1D [42], 2D [43], following a ballistic trajectory [44] or via an intermittent potential [38,45,46]).

These models can provide in some cases analytical solutions for the distribution of animals around the center, and in some cases even for the time needed to discover a target (mean first passage time, MFPT) [47–50]. For instance, the resetting rate can be tuned in order to minimize MFPT ([42,51–53], see [54] for generalization to any dimension and motion models, and [55] for generalizations to any resetting model).

However, these results are only valid under two strong assumptions: long times so that particles are in the diffusive regime, and low target density. Both of these assumptions are broken in realistic datasets of bumblebee exploration. Furthermore, none of these models provide trajectories that resemble those measured for bumblebees during exploration flights (Fig 1C).

## Persistent Turning Walker

We, therefore, considered the Persistent Turning Walker (PTW) model which is an extension of the Persistent Random Walker (PRW [56], see [57–59] for a previous use in modeling ants behaviour from a cognitive perspective).

While in PRW a trajectory is considered as a succession of straight moves separated by instantaneous jumps in the orientation domain, in PTW a trajectory is considered as a

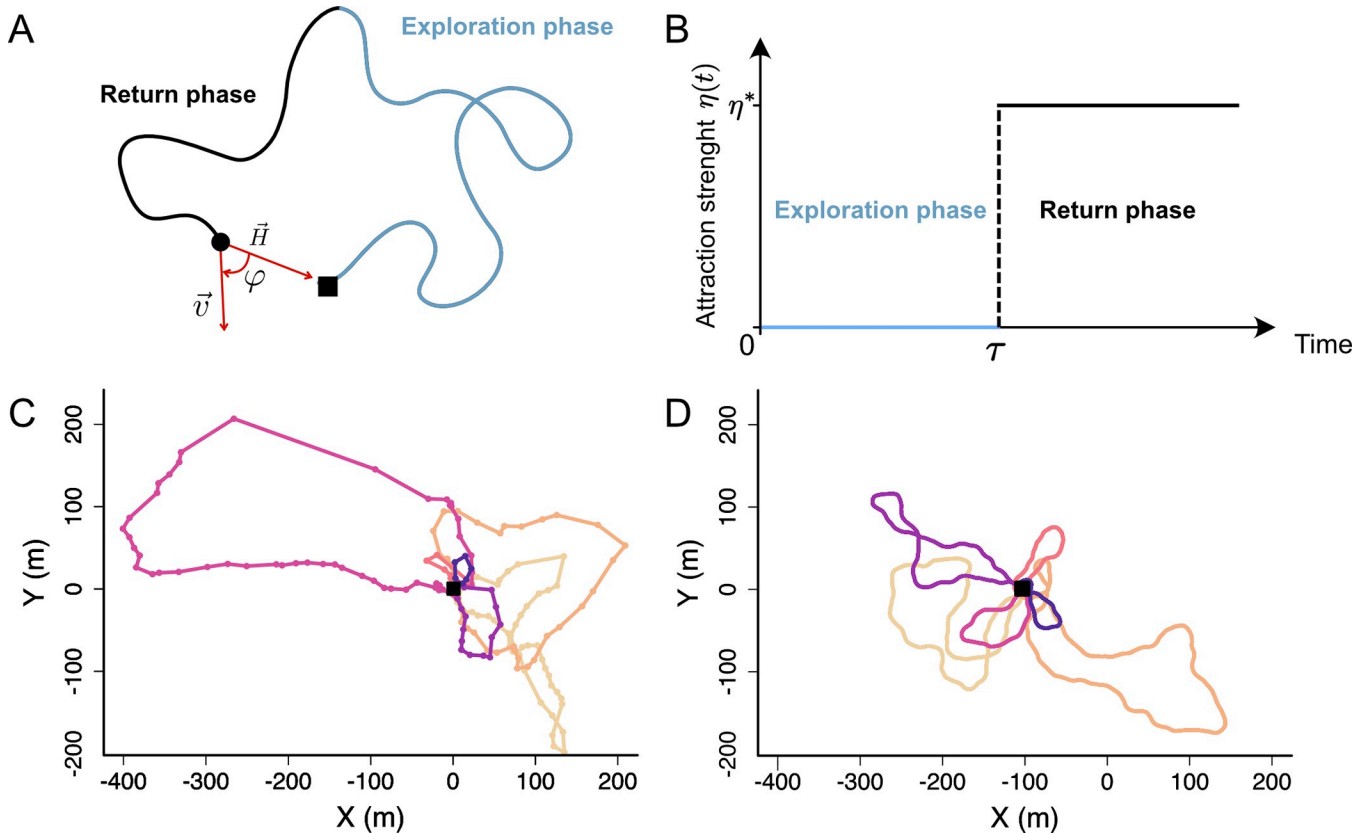

**Fig 1. Illustration of the model.** **(A)** Example of theoretical trajectory. Blue line: Trajectory during the exploration phase. Black line: Trajectory during the return phase. Black circle: bee. Black square: nest. H is the homing vector pointing towards the nest. $\vec{v}$ is the velocity of the bee. $\varphi$ is the angle between $\vec{v}$ and $\vec{H}$. **(B)** Evolution of the return strength ($\eta$) over time. At time = $\tau$, $\eta$ switches from 0 (no attraction) to $\eta^*$. **(C)** Example of an experimental trajectory [71]. Each dot represents the position of a bee recorded by a harmonic radar approximately every 3s. Different colors represent different flight loops around the nest. The sequential order of the loops is represented by the color gradient where the first loops have lightest colors (yellow to purple). **(D)** Same as C, but for a simulated trajectory with parameters $\gamma = 1.0\ s^{-1}$, $\sigma = 0.37$ rad/$s^{1/2}$, $p_{\text{return}} = 1/30\ s^{-1}$ and $\eta^* = 0.2\ s^{-1}$.

succession of moves with constant curvatures, separated by instantaneous jumps in the curvature domain.

Such a description allows for a continuously defined velocity with time-correlation of the curvature. A stochastic version has been used to model fish motion behaviour [60,61].

The PTW model describes motion at constant speed, where the heading is driven by an Ornstein-Uhlenbeck process acting on the turning speed. In free-range condition, it yields a diffusion process on large time scales [62]. It has been used recently to model an active directional filament in 2D free-range conditions [63], and some large-time properties (NESS and MFPT) have been derived when in presence of a steering potential acting upon the heading [64].

Since it yields trajectories that are similar to our data, we used the PTW model as a basis for our own model.

To confine trajectories around the nest, we added an intermittent steering potential acting on the turning speed and that activates when bees decide to return to the nest. Thanks to this steering potential, our model describes the full exploration trips of bees (starting and ending at the nest location), which facilitates comparisons between model simulations and experimental data.

## Results

### Description of the model

For the sake of simplicity, here we modelled bee movements in 2D, neglecting altitude. We assumed that bees fly at constant speed v and with varying angular speed, $\omega(t)$ (signed turning rate of the heading, measured in radians per second) which is governed by

$$d\omega(t) = -\gamma[\omega(t) - \omega^*(t)]dt + \sigma dW(t), \tag{1}$$

where $\gamma$ is an auto-correlation coefficient and $\sigma dW(t)$ introduces a gaussian noise, governed by a Wiener process [65]. The two terms of Eq 1 have opposing effects: The first term pushes the angular speed towards a target angular speed, $\omega^*(t)$, with a strength controlled by the auto-correlation coefficient $\gamma$. The second term introduces noise in the angular speed making bees change direction. Therefore, high values of $\gamma$ and low values of $\sigma$ lead to smoother and more predictable trajectories. Setting $\omega^*(t) = 0$ leads to a trajectory with no preferred direction, whose angular speed changes smoothly around zero. This is the simplest condition, resembling a diffusive process in which the animal moves aimlessly and gets further and further from its initial position as time goes by [62,63,66–69].

We modelled central place foraging by adding an attraction component to the model in order to make bees return to the nest after a certain amount of time. To implement the return to the nest (homing) we assumed that bees can locate the direction of their nest at any time using path integration (i.e. navigational mechanism by which insects continuously keep track of their current position relative to their nest position [70]), and define a homing vector, $\vec{H}(t)$ that points towards the nest [13]. Then, we assumed that the bee tries to target the angular speed that will align its trajectory with the homing vector, so we modeled the target angular speed as

$$\omega^*(t) = \eta(t)\varphi(t), \tag{2}$$

Where $\varphi(t)$ is the angle between the bee's velocity $\vec{v}(t)$ and the homing vector $\vec{H}(t)$ (Fig 1A), and $\eta(t)$ is the attraction strength that controls a switch between the exploration and return phases: During the initial exploration phase we make $\eta(t) = 0$, so that bees explore randomly and distance themselves from the nest, while during the return phase we make $\eta(t) = \eta^* > 0$, so that the bee has a continuous tendency to turn towards the nest. We assumed that bees switch instantaneously between the exploration and return phases, so

$$\begin{cases} \eta(t) = 0 \; if \; t < \tau \\ \eta(t) = \eta^* \; if \; t \geq \tau \end{cases} \tag{3}$$

where $\tau$ is the time at which the switch happens (Fig 1B). This switch may happen at any time, with a constant probability per unit of time, $p_{\text{return}}$. This means that the switching times are exponentially distributed, with an average time of $1/ p_{\text{return}}$.

The model therefore has four main parameters: The auto-correlation ($\gamma$) and the randomness ($\sigma$) control the characteristics of the flight, while the probability per unit of time to return ($p_{\text{return}}$) and the strength of the attraction component ($\eta^*$) control the duration of each exploration trip. Here we have described the continuous version of the model, but to implement it numerically we discretized it in finite time steps (see Methods).

### Calibration with experimental data

In principle, our model can describe search movements of any central place forager. Here we explored its properties focusing on a model species for which we had access to high-quality

experimental data: the buff-tailed bumblebee *Bombus terrestris*. We used the dataset of Pasquaretta et al. [71] in which the authors used a harmonic radar to track 2D trajectories during exploration flights of bees in the field. Bees carrying a transponder were released from a colony nest box located in the middle of a large and flat open field, and performed exploration flights without any spatial limitation. The radar recorded the location of the bees every 3.3s over a distance of ca. 1km (Fig 1C). In these experiments the bees were tested until they found artificial flowers randomly scattered in the field. We used 32 tracks from 18 bees.

To quantify the experimental trajectories, we first divided tracks into flight "loops", each loop being a segment of trajectory that starts and ends in the nest (Fig 1C). This extraction yielded 207 loops. We then computed four observables for each loop (Fig 2):

- Loop length: Total length of the trajectory for a given loop (Fig 2A).

- Loop extension: Maximum distance between the bee and the nest for a given loop (Fig 2B).

- Number of intersections: Number of times the loop intersects with itself (Fig 2C).

- Number of re-departures, where a re-departure is defined as three consecutive positions such that the second position is closer to the nest than the first one, but the third is again further away than the second. These events indicate instances in which the bee seemed to be returning towards the nest and turned back (Fig 2D).

We extracted these four parameters from each loop and found substantial variability in all of them (Fig 2, black lines). We then used this information to find the optimal model parameters, aiming to describe not only the average value of each observable, but also their distributions. To do so, we performed simulations covering exhaustively all relevant combinations of

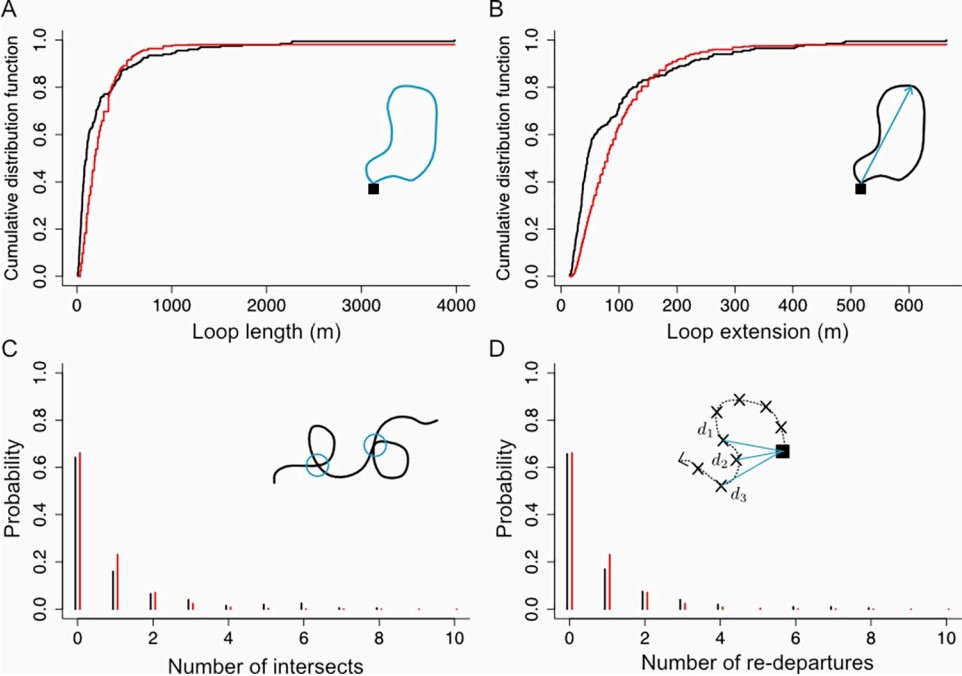

**Fig 2. Distributions of the four observables, for experimental and simulated data.** Black lines: experimental data. Red lines: model predictions using the optimal $\gamma = 1.0\ s^{-1}$, $\sigma = 0.37$ rad/s$^{1/2}$, $p_{\text{return}} = 1/30\ s^{-1}$ and $\eta^* = 0.2\ s^{-1}$. Insets: Schematic of each observable. **(A)** Cumulative distribution function of loop lengths for our full dataset. **(B)** Same as A, but for the loops extension. **(C)** Probability distribution of the number of trajectories intersects per 100m traveled. **(D)** Same as C, but for the number of re-departures per 100 m traveled.

our four parameters. For each combination of parameters, we simulated 1000 loops, extracted the distributions for the four observables, and chose the parameter combination that best approximated the experimental distributions for the four observables (see Methods). This procedure resulted in the optimal parameters $\gamma = 1.0\ s^{-1}$, $\sigma = 0.37$ rad/$s^{1/2}$, $p_{\text{return}} = 1/30\ s^{-1}$ and $\eta^* = 0.2\ s^{-1}$, which give a good approximation to the experimental distributions of observables (Fig 2, red lines), and trajectories that qualitatively resemble the experimental ones (Fig 1D).

Note that our calibration procedure yields a switching time from the "exploration phase" to the "return phase" of 30 seconds on average. This may be an underestimation since the radar can detect targets at a maximum distance of about 1km [72] and bumblebees can fly beyond this limit depending on forage availability in the landscape [73]. However, to the best of our knowledge, these are so far the best data available.

## Model predictions

**Attraction to the nest limits the exploration range of bees.** An unrealistic feature of existing diffusive models is their long-term behaviour: If given enough time, the forager reaches extremely far distances with respect to the nest, never returning to it. To illustrate the impact of central place foraging on the simulation of bee exploration range, we compared our model with attraction to the nest to an alternative one in which the attraction is absent (i.e., making $\eta^* = 0$ in Eq 3). We simulated 1000 trajectories with each model for different amounts of time, and studied how the distribution of bees around the nest changes over time. As expected, attraction retains bees tightly localized around the nest (~250m) (Fig 3A and 3B, blue). More interesting, it makes the distribution of bees stationary: In a model without attraction, bees constantly wander away from the nest, and their distribution depends on how much time we allow for the bee to explore, becoming wider as time goes by (Fig 3B and 3C, orange). In contrast, the attraction component makes the forager return to the nest periodically, so the distribution remains stationary once the forager has had enough time to perform more than one loop on average (Fig 3D, blue).

This demonstrates the importance of taking into account the attraction component when modelling bee movements (instead of considering a diffusive model) to be able to reproduce realistic trajectories.

To highlight the effect of behavioural parameters upon the spatial extent covered by exploration, we present in S1 Fig the marginal effect of the three main parameters ($\alpha$, $\eta^*$ and $\sigma$) on the range explored by the bee, as measured by the long-time Mean Square Displacement (MSD), i.e. the variance of the stationary distribution (also known as the Non-Equilibrium Stationary State, or NESS, of the motion process in Statistical Physics).

**Distant flowers are more often discovered in low-density environments.** We estimated the probabilities of flowers to be discovered (and thus potentially pollinated) by bees in a simulated field characterized by a random and uniform distribution of flowers, an average density of $1.3\ 10^{-4}$ flowers/$m^2$ and a diameter of 70cm (for the sake of simplicity here a "flower" is equivalent to a feeding location, which may be a single flower or a plant containing several ones). We assumed that a flower was discovered by a bee whenever its distance to the bee's trajectory was below a threshold, given by the bee's visual perception range (see Methods). We focused on vision rather than olfaction because it is the main sense that bees use to accurately navigate the last meters towards a particular flower, while olfaction is used at a broader spatial scale [74]. Using these conditions, we simulated 1000 foraging trips, each of them lasting 900 seconds, and for each flower we computed the probability to be found in a given trip (i.e., the proportion of simulations in which the trajectory overlaps with the flower's area of attraction). This probability falls exponentially with the distance between the flower and the nest (Fig 4A, red line).

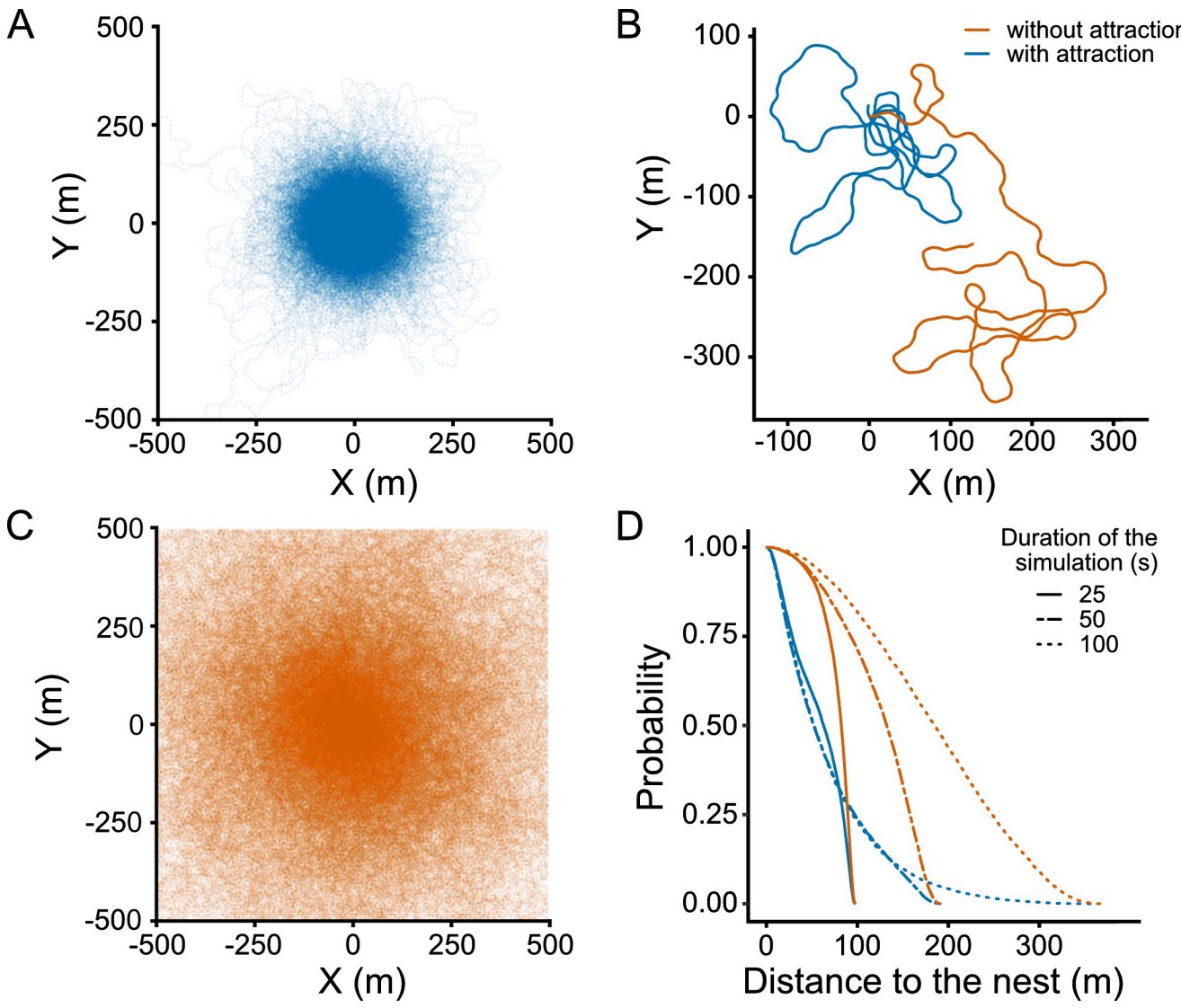

**Fig 3. Probability of presence of a bee around the nest.** (**A**) Overlay of 1000 trajectories with attraction to the nest ($\eta^* = 0.2\ s^{-1}$) simulated during 900 s. (**B**) Example trajectories with and without attraction, simulated during 500 s. The nest is located at (0,0). Blue: model with attraction. Orange: model without attraction. (**C**) Same as A but without attraction to the nest ($\eta^* = 0$). (**D**) Probability to find a bee below a given distance to the nest (i.e., inverse cumulative probability distribution for the distance to the nest) after different amounts of time. Blue: model with attraction (stationary distribution of bees). Orange: model without attraction (non-stationary distribution of bees).

We then tested the influence of a potential perceptual "masking effect" in which the probability of discovering a flower does not only depend on its distance to the nest, but can also be influenced by the presence of other flowers around it (Fig 4B). This dependence exists because a bee that finds a flower might not continue its trajectory, but might rather stop to collect nectar. Once nectar collection is over, the bee may continue exploring, but after visiting a few flowers the bee returns to the nest to unload its crop. For example, in a scenario where there are just two flowers equidistant to the nest, both flowers should be visited equally. However, if another flower is added, it can capture visits that would otherwise visit one of the original flowers, reducing the probability that it is discovered (Fig 4B). For the sake of simplicity, we assumed that each bee returns to the nest after discovering a single flower. The first qualitative

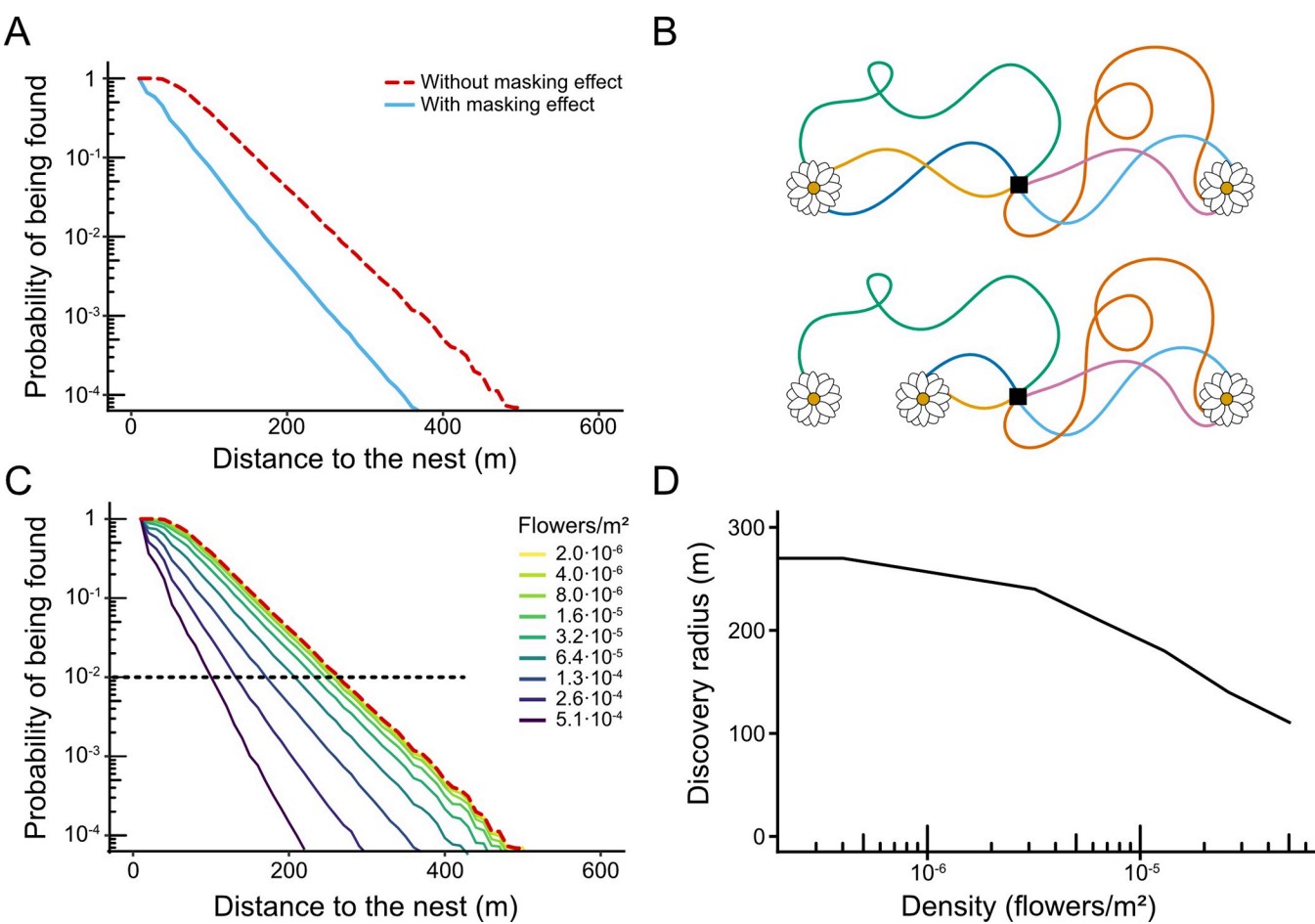

**Fig 4. Probability that flowers are discovered.** (A) Probability that a flower is found as a function of its distance to the nest. We simulated exploration trips in a field of uniformly distributed flowers with density 1.3 $10^{-4}$ flowers/m$^2$ and flower size 70 cm. For each flower, we computed the probability that it was found in each exploration trip, and we show this probability as a function of the distance between the flower and the nest. Results were computed over 6000 simulated trips of 900s in 10000 environments for each density (negligible bar errors not reported). Red line: Probability calculated without taking into account the masking effect. Blue line: Probability calculated taking into account the masking effect (i.e., only counting the first flower that was discovered in each trip). **(B) Illustration of the masking effect.** The probability of discovering a flower depends on the presence of other flowers. In a scenario where there are just 2 flowers equidistant to the nest, both flowers should be visited equally (top). However, if another flower is added, it can capture visits that would otherwise visit one of the original flowers (bottom). Black square: nest. **(C)** Same as (A), but for different flower densities. Red dotted line: Probability calculated without taking into account the masking effect. This probability is independent of the density of flowers. Solid lines: Probability calculated taking the masking effect into account. Black dotted line: threshold probability at which we consider an area that has a high probability of being pollinated. **(D)** Radius of the area around the nest that has a high probability of being discovered (i.e., where the probability that flowers are discovered is above $10^{-2}$) as a function of flower density.

consequence of the masking effect is to reduce the probability that flowers distant to the nest are discovered (Fig 4A, blue). The second consequence is that it can introduce a dependence of flower density on discovery rate. In the absence of masking, only two factors determine the probability that a flower is discovered: its size (which determines the distance from which it can be perceived) and its distance to the nest. In contrast, when the masking effect is taken into account, the number of discoveries visits also depend on the overall density of flowers in the environment, falling more sharply with distance when this density is higher (Fig 4C).

This dependence with flower density implies that the area around the nest where flowers have a high probability of being discovered depends on flower density. To estimate the size of this area, we set a threshold at a probability of $10^{-2}$ per trip (black dotted line in Fig 4C), and computed the "discovery radius" as the distance at which flowers' probability of being

discovered remains above this threshold. At low flower densities, the discovery radius reaches 270 meters, and is limited by the bees' exploration range (i.e., their tendency to return to the nest after a certain time, even if no flowers have been found; compare this radius with the distribution in Fig 3C). Due to the masking effect, the discovery radius decreases as flower density increases (Fig 4D).

**Populations of bees find more flowers at intermediate densities.** We then explored the influence of the masking effect on the total number of flowers discovered by a population of bees (i.e. a bumblebee colony).

To study this effect, we computed the total number of flowers discovered by a bee colony as a function of density and flower size. We considered a field with flowers of a given size uniformly and randomly distributed with a given flower density, simulated 100 exploration trips, and counted the number of flowers that were discovered at least once. When we performed this simulation neglecting the masking effect (i.e., assuming that a bee discovers all the flowers that intersect with its trajectory, not being affected by previous discoveries), we found that the number of flowers discovered increased with flower density and flower size, as these factors make flowers more plentiful and easier to find (Fig 5, dashed lines). However, the masking effect reverses this trend (Fig 5, solid lines): For low densities, the masking effect is weak and the number of discovered flowers increases with density, but at high flower densities, bees become "trapped" around the nest by the flowers immediately surrounding it, which accumulate most of the visits. Therefore, there is an optimum density that results in the highest number of different flowers discovered. Since the masking effect is stronger for larger flowers, the effect of size also reversed, with the number of discovered flowers decreasing as flower size increases (Fig 5, solid lines).

These results are robust to the assumptions of the model. Firstly, they do not depend on the precise number of flowers discovered by each bee in each exploration flight. In many natural conditions, bees may need to discover and visit several flowers to fill their nectar crop to capacity before deciding to return to the nest. Taking these multiple discoveries into account (1 to 8 discoveries) leaves our results qualitatively unchanged (Figs 5B and S2).

Our results are also robust to nectar depletion. Once a flower is visited by a bee, its nectar load may be partially depleted. The next bee visiting the flower may therefore be less inclined to terminate its exploration flight and return to the nest after visiting this flower, so the number of flowers discovered by each bee before returning to the nest may depend on the previous exploration flights performed by other bees in the colony. To account for flower depletion, we ran a simulation in which bees will ignore any flower that has already visited in a previous exploration flight. In this case, the maximum at intermediate densities is lost, with higher flower density always leading to more discovered flowers (S3E Fig). However, for large patch sizes, we know that flower depletion is only partial and visiting each flower has a cost in terms of time and energy (bees must land on each flower, even if it is depleted). To account for this cost, we limited the total number of depleted flowers a bee will visit before returning to the nest, and in this condition we again observe a maximum of discovered flowers at intermediate densities (S3A–S3D Fig).

## Discussion

How pollinators search for flowers is of fundamental importance for behavioural research, pollination and conservation but remains poorly quantified. Here we developed a realistic model of bee search movements based on their observed tendency to make exploratory loops that start and end at their nest location. We used a Persistent Turning Walker model which has inspired some developments in other animals, especially fish [41–43] as well as in robotics [44]

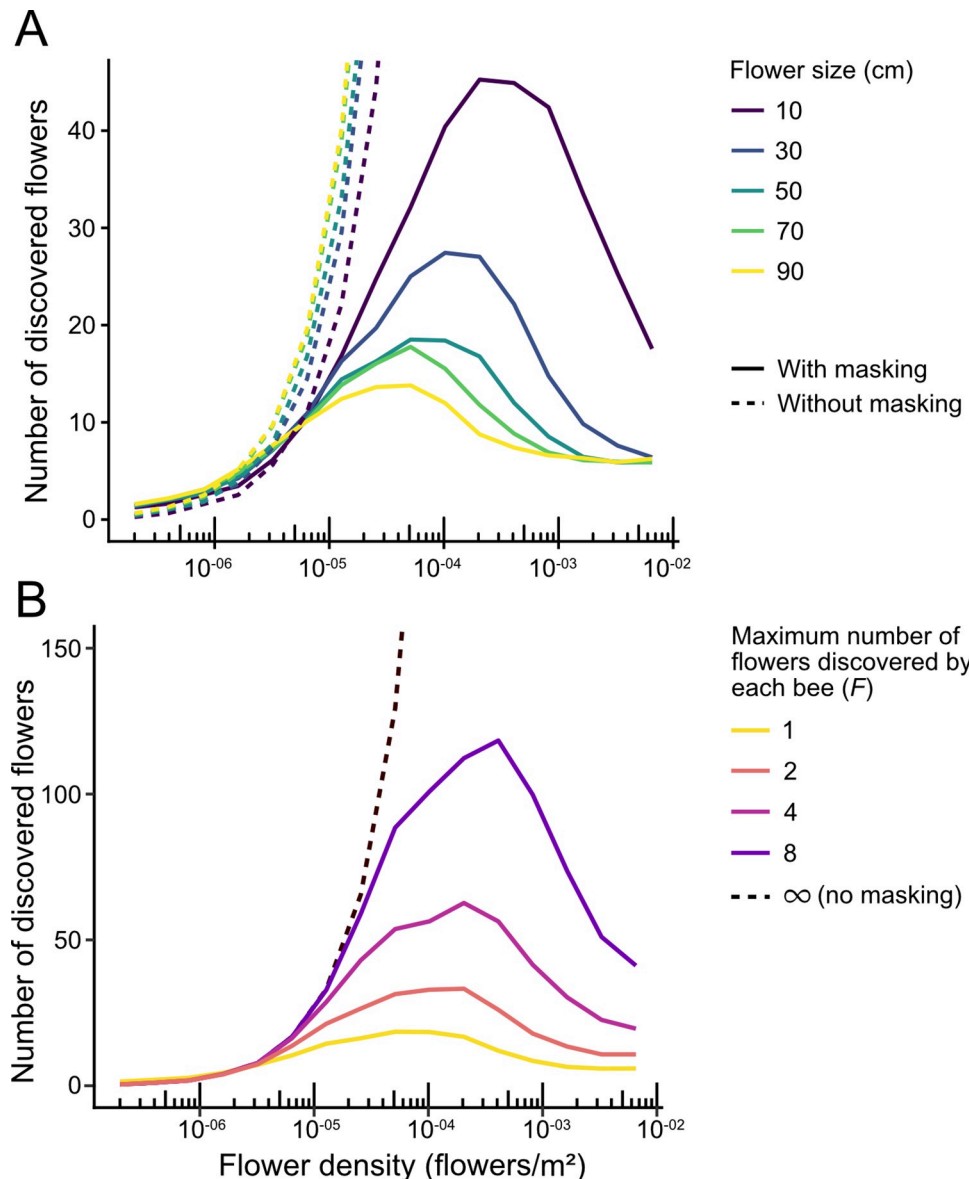

**Fig 5. Number of different flowers discovered by a group of bees as a function of flower density. (A)** Number of different flowers discovered in 100 exploration trips of 900 s, in an environment with randomly distributed flowers. Results are averaged over 80 simulations, keeping the environment stable for every simulation. Solid lines: Number calculated taking into account the masking effect (i.e., only counting the first flower that was discovered in each trip). Dotted lines: Results without taking into account the masking effect. **(B)** Same as A but for a given flower size (50 cm), and assuming that each bee will return to the nest only after having discovered a number of flowers ($F$) (note that box A corresponds to $F = 1$ for the simulations with masking effect, and $F = \infty$ for the simulations without masking effect). Line colors represent the maximum number of flowers discovered by each bee ($F$).

where it has been proven to display better coverage properties than classical random walks [45]. Our model, calibrated with real bee behavioural data (i.e. bumblebee radar tracks), produces two-dimensional trajectories with progressive changes of direction driven by the continuous evolution of the angular velocity $\omega(t)$. Using this approach, we documented a neglected yet potentially fundamentally important effect for flower discovery by bees: a perceptual

masking effect that influences the probability of bees to find flowers not only based on their size and spatial location, but also on the presence and characteristics of other flowers around them.

Previous models assume that bees explore their environment randomly using Lévy flights or other diffusive processes [12,21,22]. In a diffusive model, individuals are able to wander away from the nest indefinitely if given enough time. In contrast to these models, our model replicates looping trajectories observed in real bees [14,29], which confines the presence of individuals around a nest (Fig 3). As a consequence of the periodic returns of bees to the nest, their distribution becomes independent of the time given to explore. This result has the important consequence that, under the assumptions of our model, longer simulation durations result in a more thorough exploitation of the foraging area around the nest, but not in a larger area.

By explicitly simulating individual trajectories in complex environments, our model revealed how the presence of a flower may decrease the probability of discovering another, a phenomenon that we call "perceptual masking effect". Although in this study we strictly focused on flower discovery during exploration flights, our results suggest that the masking effect can have broader influences on exploitation patterns, bee foraging success and plant pollination. Indeed, at the level of individual bees, flower discovery can impact site fidelity by foragers and their tendency to develop traplines to regularly revisit known feeding locations [17]. Given the perceptual masking effect, flower patches may not be discovered with the same probability or in the same order, which may impact their likelihood to be exploited and the ability for bees to develop efficient routes minimizing overall travel distances between different flowers [21]. The masking effect may also influence the global foraging success of the colony, which depends on the number of flowers discovered collectively by all the bees of a colony, because a flower discovered and exploited by a bee will be at least partially depleted, giving marginal benefit to later visitors. For this reason, what counts is not the total number of visits that bees perform, but rather the total number of different flowers discovered by the colony. In our simulations, colonies tended to find more flowers when they were small and at medium densities (Fig 5). This suggests that there is an optimal flower size and density at which collective foraging efficiency is optimized (although the effect of size on foraging efficiency will be compounded with the greater reward provided by bigger flowers on average).

Similar extrapolations of our results on first flower discovery can be made regarding pollination. At the plant level, we found that flowers distant to the nest were more often visited in low density environments (Fig 4). Since bees disseminate pollen (and thus mediate plant sexual reproduction) when visiting flowers, this may generate lower probabilities of pollination at high flower densities, by which the area that is pollinated around the nest decreases as the density of flower increases. If this prediction is verified in future studies, this would mean that the overall distribution of flower patches directly impacts their pollination and should be taken into account when designing strategies for crop production and assisted pollination.

Our search model is a scaffold for future quantitative characterization of the movement of bees, or any central place forager, across time and landscapes. Although we limited our study to flower discovery probability, and therefore only provided predictions for first flower discovery, the model could be used to investigate the full foraging trips of bees, and how they change through time as bees acquire experience with their environment and develop spatial memories [13]. It would be particularly interesting to integrate this exploration model into existing learning exploitation models proposed to replicate route formation by bees [20–22] and to study dynamics of resource exploitation by populations of bees [22,75]. Once a flower is discovered, its location can be learned, and new exploration may start from it, ultimately allowing for the establishment of traplines. This would be modelled via a modification of the attraction component, which can be modified to point towards previously-discovered flowers instead of the

nest. Importantly, such model predictions (flower discovery probability, visitation order, flight trajectories) can be experimentally tested and the model calibrated for specific study species. This will greatly facilitate improvement and validation for potential applications. As discussed above, robust predictive models of bee movements including both exploration and exploitation would be particularly useful for improving precision pollination (to maximize crop pollination), pollinator conservation (to ensure correct population growth and maintenance), but also in ecotoxicology (to avoid exposure of bees to harmful agrochemicals) and legislation (to avoid unwanted gene flow between plants). Beyond pollinators, our minimal persistent turning walker model could be calibrated to apply to a wide range of species, providing a data-based quantification and predictions for further exploration of the broader interactions between central place foraging animals and their environment.

## Methods

The codes used to perform all the simulations, data analyses and figures are available in the S1 Data.

### Modeling nest and flower detection

Bumblebees can detect an object when it forms an angle of 3˚ on the retina of their compound eyes [76]. Therefore, for every model simulation, we set the flowers' size and calculated the distance at which the bees are able to detect them. We call this the "perception distance". We considered that a bee discovered a flower when it was located at a distance to the bee inferior to the perception distance. We did not take into account the olfactory perception since it could be less reliable because of other factors like wind direction and the flower's species. If taken into account, this would only impact the perception distance of the flowers and the results would not be qualitatively different.

### Analysis of experimental data

We used the dataset of Pasquaretta et al. [71] in which the authors tracked exploratory flight trajectories of bumblebees in the field with a harmonic radar. Bees carrying a transponder were released from a colony nest box located in the middle of a large and flat open field, and performed exploration flights without any spatial limitation. The radar recorded the location of the bees every 3.3s over a distance of ca. 800 m, and with an accuracy of approximately 2 m [14]. The bees were tested until they found one of three 20-cm artificial flowers randomly scattered in the field. The position of these flowers was changed whenever one of them was found to prevent the bees from learning their location, but their presence may still affect the bees' trajectories. We first attempted to control for this factor by removing all trajectories where bees passed near an artificial flower, but this introduced a significant bias towards short trajectories, because bees are less likely to find a flower when they stay near the nest. Therefore, we used the full dataset, and in order to remove the effect of the bees hovering around and exploiting the artificial flowers, we summarized all the points detected in an area within 6 m of an artificial flower as a single point at the location of the flower. This threshold of 6 m was derived from a 4 m perception distance corresponding to 20 cm flowers, plus 2 m to account for the experimental noise. All trajectories are given in S2-section V.

### Dividing trajectories into loops

In order to quantify the trajectories, we divided the tracks into "loops". We defined a loop as a fragment of a trajectory that starts when the bee leaves that nest and finishes when it enters

back. The colony nest box used in the experiments was rectangular, with a diagonal of 37 cm, meaning that the bees were able to see it at approximately 7m. However, in this case we set a higher threshold of 13 m to avoid including learning flights (i.e., flights during which the bee makes characteristic loops to acquire visual memories of target locations such as the nest for navigation [70]) into the set of exploratory data. While our model does not produce learning flights, for consistency we also used the 13-m radius around the nest in our simulations.

## Model simulations

All simulations start at the nest (which is located at position 0,0), with a random initial heading, and with zero angular velocity. To simulate the trajectories, we discretized the model using a time step $\Delta t = 0.01$. Therefore, at every step we calculated position $\vec{x}(t + \Delta t)$ with:

$$\vec{x}(t + \Delta t) = \vec{x}(t) + \vec{v}(t)\Delta t. \tag{M1}$$

Then, the direction $\theta(t + \Delta t)$ was calculated using

$$\theta(t + \Delta t) = \theta(t) + \omega(t)\Delta t. \tag{M2}$$

The velocity $v(t + \Delta t)$ was calculated with

$$\vec{v}(t + \Delta t) = v \begin{pmatrix} cos(\theta(t + \Delta t)) \\ sin(\theta(t + \Delta t)) \end{pmatrix}, \tag{M3}$$

where $v$ is the speed, which is a constant in our model.

Lastly, we calculated the angular speed $\omega(t + \Delta t)$. For this, we used the Green function for Ornstein-Uhlenbeck processes over $\Delta t$ (see [60] for details), obtaining

$$\omega(t + \Delta t) = \omega(t)e^{-\gamma\Delta t} + \omega^*(1 - e^{-\gamma\Delta t}) + \varepsilon, \tag{M4}$$

where $\omega^*$ is the target angular speed (governed by Eqs 2 and 3), and $\varepsilon$ is a random number governed by a Gaussian distribution with mean 0 and variance

$$s^2 = \sigma^2 \frac{1 + e^{-2\gamma\Delta t}}{2\gamma}. \tag{M5}$$

## Flower discovery models

Our basic model with masking effects assumes that a bee returns to the nest after discovering one flower and will ignore any flowers encountered on its way back. We implemented this by simply stopping our simulation when the bee discovers the first flower (if no flowers are discovered, the simulation continues until the central-place-foraging component makes the bee return to the nest).

Our multiple-discovery model (Figs 5B and S3) assumes that a bee will continue its original trajectory after each discovery (i.e. for simplicity we do not include the landing on the discovered flowers), and will return to the nest after discovering $F$ flowers (so we stop the simulation when the bee discovers the $F$-th flower).

Our model with flower depletion (S4 Fig) takes into account the order in which bees make their exploration trips. If a flower has been discovered by a bee in a previous trip, it's marked as depleted. We stop the simulation when a bee discovers a non-depleted flower, or after discovering a maximum number $F_{depleted}$ of depleted flowers.

## Parameter fitting

In order to fit the parameters of the model, we explored systematically all relevant combinations within the relevant range for each parameter. To do this more efficiently, we transformed one of the four parameters of the model into a more tractable one: We substituted the variance of the noise introduced by the Wiener's process ($\sigma$) for the variance of the angular speed $\omega$, which has a more direct impact on the experimental data. These two variables are related by [60]:

$$Var(\omega) = \Omega = \frac{\sigma^2}{2\gamma}.$$

We also defined

$$\alpha = \frac{1}{p_{return}}.$$

After these transformations, our model is defined by the four parameters ($\gamma, \Omega, \alpha, \eta^*$). To find the optimal values of these parameters, we exhaustively explored the 4-dimensional parameter space in the relevant range of each parameter: We explored all 6160 different combinations resulting from the following values of each parameter:

- $\gamma \in (0.5, 0.6, 0.7, 0.8, 0.9, 1.0, 1.1, 1.2, 1.3, 1.4, 1.5)$

- $\Omega \in (0.01, 0.03, 0.05, 0.06, 0.07, 0.08, 0.09, 0.1, 0.125, 0.15)$

- $\alpha \in (10, 20, 25, 30, 35, 40, 50)$

- $\eta^* \in (0.05, 0.1, 0.15, 0.2, 0.25, 0.3, 0.35, 0.4)$

For each combination of parameters, we simulated $10^3$ loops and computed the distribution of each of the four observables defined in Fig 2. Then we computed the distance between the experimental distribution of each observable and the simulation results: for the two continuous observables (loop length and extension), this distance was computed as the area between the observed cumulative distribution function and the simulated one. For the two discrete observables (numbers of self-intersection and re-departures), the distance was computed as the sum of the absolute differences between all points of the two probability distributions. This yielded four distributions of distances over the 6160 combinations. Since the four observables are heterogeneous (two are continuous measures, two are discrete), we had to re-normalize the distances to ensure that each observable is given the same weight. We did this by translating the distances into their quantile in their corresponding cumulative distribution (e.g., a distance translated into 0.12 means that it is within the lowest 12%). Finally, we retained the combination that yielded the lower quantile averaged over the four observables (see S1 Text, section 2).

The best parameters combination was found to be: $\gamma = 1.0 \ s^{-1}$, $\Omega = 0.07 \ \text{rad}^2/\text{s}^{-2}$, $\alpha = 30 \ s$ and $\eta^* = 0.2 \ s^{-1}$. It corresponds to the marginal local minima for the four observables (see S1 Text, section 2). Simulated trajectories closely resemble data trajectories (Fig 1B), and the model is able to produce loops with an elongated shape, as well as a diversity of loop lengths.

This unique set of parameters assumes that all bees are identical, while in reality inter-individual differences exist (S2 Fig), for example due to differences in age, experience, learning or size [77,78]. However, each bee can display a large diversity of loop parameters, covering a similar range as the overall population (S1 Fig). We therefore considered that separate fits for each individual were not justified. The fact that our model reproduces not only the mean but also the variability of the four observables we defined (Fig 3) supports this choice.

## Supporting information

**S1 Fig. Sensitivty of the Mean Square Displacement (MSD) to the behavioural parameters.**
We numerically computed the MSD ($m^2$), varying each parameter in turn, leaving the others
unchanged (keeping their values fitted from the dataset). The MSD was estimated using 10^5
simulated loops for each point, using the default parameters $\gamma = 1.0$ $s^{-1}$, $\sigma = 0.37$ rad/ $s^{1/2}$, $p_{re-turn} = 1/30$ $s^{-1}$ and $\eta^* = 0.2$ $s^{-1}$ and varying (A) $\alpha$ ($s$), (B) $\eta^*$ ($s^{-1}$) $or$ (C) $\Omega = \frac{\sigma^2}{2\gamma}$ (rad$^2$/s$^{-2}$) while
leaving the other parameters unchanged. Note that for smaller values of $\eta^*$, the distribution
tends to (non stationary) diffusion, so, some loops are censored at one hour. The effect of
$\alpha = \frac{1}{p_{turn}}$, the mean duration in exploration mode, appears quite linear, which is not surprising
but would call for an analytical demonstration. During the return phase, $\eta^*$ represents the
intensity of the steering (the potential stiffness), since for smaller values of $\eta^*$, the relaxation to
the preferred turning speed would be less effective. As $\eta^*$ gets smaller and smaller values, the
steering vanishes, so that the bee would adopts a diffusive behavior, with no Non Equilibrium
Stationary State (NESS). On the other hand, we observe a clear effect of saturation for large
enough values of $\eta^*$, meaning that the effectiveness of the steering is limited by the relaxation
time gamma. Finally, $\Omega$ controls the level of noise the turning speed can undergo. For $\Omega = 0$,
the turning speed has no noise at all, and the process becomes deterministic: bee would fly
from the nest ballistically in the exploration phase and go back ballistically to the nest after
turning maneuver induced by the steering process. In this case, given the initial condition of
null turning speed, the trajectories would push bees the farthest from the nest during explora-
tion (hence, maximal MSD), while larger values of $\Omega$ would drive bees to meander around the
nest, leading to trajectories that remain closer to the nest.
(TIFF)

**S2 Fig. Variability of each observable across individuals in the experimental dataset. (A)**
Loop lengths (m) for each bee, as defined in Fig 3 in the main text. Boxplots, show the median
(middle line), 25 and 75% quantiles (box), range of data within 1.5 interquartile deviations
(whiskers), and outliers (dots). **(B)** Same as A but for the loop extension (maximum distance
between the nest and the individual). **(C)** Same as A, but for the number of re-departures per
100m traveled. A re-departure is defined as three consecutive positions such that the second
position is closer to the nest than the first one, but the third is again further away than the sec-
ond. **(D)** Same as A but for the intersections (number of times the loop intersects with itself).
(TIFF)

**S3 Fig. Number of different flowers discovered by a group of bees as a function of flower
density, when bees discover more than one flower per trip. (A)** Number of different flowers
discovered in 100 exploration trips of 900 s, in an environment with randomly distributed
flowers. Results are averaged over 80 simulations, keeping the environment fixed for every
simulation. Solid lines: Number calculated taking into account the masking effect (i.e., only
counting the first flowerF = 2 flowers that wereas discovered on each trip). Dotted lines: Proba-
bility calculated without taking into account the masking effect. **(B)** Same as (A), but for $F = 4$.
**(C)** Same as (A), but for $F = 8$. Note the difference of scales for the ordinates.
(TIFF)

**S4 Fig. Number of flowers discovered by a group of bees as a function of flower density,
when accounting for flower depletion.** Here we assumed that an individual will continue
exploring after visiting an already-explored flower, and will return to the nest only when
encountering a fresh flower or after a fixed number of visits to already-visited flowers ($F_{depleted}$). **(A)** Number of different flowers discovered in 100 exploration trips of 900 s, in an

environment with randomly distributed flowers, and with $F_{\text{depleted}} = 1$ (note that this value of $F_{\text{depleted}}$ makes the simulation identical to that in the main text). Results are averaged over 80 simulations, keeping the environment fixed for every simulation. Solid lines: With masking effect. Dotted lines: Without masking effect (the bee does not react to previous flower encounters, and all discovered flowers were counted). Colors correspond to different flower sizes. **(B)** Same as A but with $F_{\text{depleted}} = 2$. **(C)** Same as A but with $F_{\text{depleted}} = 4$. **(D)** Same as A but with $F_{depleted} = 8$. **(E)** Same as A but with $F_{\text{depleted}} = \infty$. **(F)** Same as A but for a given flower size (50 cm), with colors representing the value of $F_{\text{depleted}}$.
(TIFF)

**S1 Text. Raw results and figures.**
(PDF)

**S1 Data. Data and code sources for analysis and simulations.**
(ZIP)

## Acknowledgments

We thank Thibault Dubois, Tamara Gómez Moracho, Cristian Pasquaretta, Joe Woodgate, James Makinson, Joanna Brebner and Lars Chittka for sharing their data of bumblebee flight tracks using radar.

## Author Contributions

**Conceptualization:** Ana Morán, Mathieu Lihoreau, Jacques Gautrais.

**Data curation:** Ana Morán, Jacques Gautrais.

**Formal analysis:** Ana Morán, Jacques Gautrais.

**Funding acquisition:** Ana Morán, Mathieu Lihoreau, Jacques Gautrais.

**Methodology:** Ana Morán, Alfonso Pérez-Escudero.

**Project administration:** Alfonso Pérez-Escudero.

**Software:** Ana Morán, Jacques Gautrais.

**Supervision:** Mathieu Lihoreau, Alfonso Pérez-Escudero, Jacques Gautrais.

**Writing – original draft:** Ana Morán, Alfonso Pérez-Escudero.

**Writing – review & editing:** Ana Morán, Mathieu Lihoreau, Alfonso Pérez-Escudero, Jacques Gautrais.

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
