## [Decision Letter · Decision Letter 0]

1 Nov 2022

Dear Mrs Morán,

Thank you very much for submitting your manuscript "Modeling bee movement shows how a perceptual masking effect can influence flower discovery, foraging efficiency and pollination" for consideration at PLOS Computational Biology.

As with all papers reviewed by the journal, your manuscript was reviewed by members of the editorial board and by several independent reviewers. In light of the reviews (below this email), we would like to invite the resubmission of a significantly-revised version that takes into account the reviewers' comments.

In preparing your revision, please make sure to address all the issues raised by Reviewer #2, including how sensitive your results are to some model assumptions that are not very strongly supported by empirical evidence, as well as the impact/relevance of model predictions. Also, I have noticed that Reviewer #1 suggests to discuss how your results align with those in a study authored by me. Please feel free to consider that suggestion only if you find that it contributes to the discussion significantly. Otherwise, feel free to rebut this suggestion.  

We cannot make any decision about publication until we have seen the revised manuscript and your response to the reviewers' comments. Your revised manuscript is also likely to be sent to reviewers for further evaluation.

Sincerely,

Ricardo Martinez-Garcia

Academic Editor

PLOS Computational Biology

Natalia Komarova

Section Editor

PLOS Computational Biology

Reviewer's Responses to Questions

**Comments to the Authors:**

Reviewer #1: Dear Authors,

I find this to be a sound and well-designed model, and an important contribution to the body of bee models. It lays the groundwork for improvements in the modelling of bee exploratory behaviour and provides insights into some of the factors that could be affecting flower discovery beyond simple distance from the nest that is usually assumed. The manuscript provides a thorough assessment of the model, which is fitted using experimental data. I have a just few general comments that should clarify the biological insights resulting from the model and a few specific comments that should help readers to better understand the model.

General Comments:

It is my understanding that your model mimics exploratory flight behaviour in which the bees are searching for flowers and learning the landscape, rather than directed foraging flights (such as are observed by Osborne et al (2013)/reference 40 in your manuscript or Woodgate et al (2016)/reference 14 in your manuscript). Naturally, a flower has to be discovered before it can be pollinated and, as you point out, this behaviour is under-studied and models generally assume that the bees are all-knowing or that discovery has a simple relationship with distance from the nest, so it's great to see a model that examines this behaviour in more detail. However, some of your results make it sound as if you are modelling the more focused foraging behaviour. For example, in line 188, the section title is “Distant flowers are more often visited in low-density environments”. I would suggest changing this to “Distant flowers are more frequently discovered in low-density environments” or something similar, as well as going through the results section to make sure that it is clear the topic is exploratory behaviour and flower discovery. I do find that your discussion section makes this distinction clear, however.

I’m also curious as to why you chose to assume that the bee returns to the nest after discovering a single flower (or feeding location). How realistic is this for exploration behaviour? Do you think it leads to an overestimation of the masking effect? How does this behaviour fit in with exploration vs exploitation behaviour (to use the language of Woodgate et al (2016)) or the understanding that bumblebees use trapline foraging techniques?

Specific Comments:

Line 84: could you clarify what you mean by angular speed? If it includes both speed and direction, should it instead be angular velocity? Or is ω(t) only controlling the speed (frequency?) of direction changes?

Line 90: should this be ω*(t)?

Line 300: I’m not clear on what you mean by “at different scales.”

Reviewer #2: In the present manuscript the authors explore central-place foraging in bees with the help of a stochastic model. The model is essentially based on a random walk process based on a angular speed subject to a noise, together with an exploration/homing switching mechanism controlled by a constant switching rate. The parameters of the model are conveniently fitted to experimental data and the resulting numerical trajectories are used to explore the effects that target density/size have on the encounter probability between the bees and their targets (flowers).

While the starting point of the work is rather interesting (the idea that target density and/or clustering of targets may have an influence on the encounter probabilities) and the model/methodology used look reasonable, my impression after reading the manuscript is that there is nothing that this work adds on the knowledge we have about the encounter processes and/or central place foraging. To illustrate this, let me discuss on the "model predictions" that are provided in the main section of the manuscript, in their order of appearance:

i) Attraction to the nest limits the exploration range.

It is obvious that this must happen whenever a homing mechanism is introduced in the model. What the authors seem to obviate is that there is a large literature on stochastic processes where this effect has been accurately quantified in an analytical way. The probability distribution of diffusive models is well known (at least in the large-time regime), and the same is true for example for random walk models in which walkers return to the nest balistically with a given speed after a random time (see for example Masó et al. Physical Review E 100, 042104, and references there in). So, the main scaling properties of such central place mechanisms are well known (see also Pal et al. Physical Review Research 2, 043174). In case the present model can add something relevant to such known results then this should be discussed in detail, but instead the manuscript just discusses the effect of homing on the exploration range in a very qualitative and naive way.

ii) Distant flowers are more often visited in low-density environments

Again this result is obvious and trivial, so the interest of the work could be on exploring how this effect actually occurs, but again the discussion provided is mostly qualitative. It is for example very easy to justify the exponential decay found in the encounter probabilities in Figures 4a and 4c. If one defines the habitat as divided into concentric regions around the nest, the probability to reach a target at a given distance is given by the probability to cross the inner regions previously. This "crossing probability" is obviously proportional to the area occupied by the targets, and so to the target density. This means that the characteristic decays in Figures 4a and 4c will be proportional to the density (except for very low densities, as in this case targets are so sparse that that inner regions barely contain any target. In that case the characteristic decay will tend to a constant value, as it is seen for lower densities in Figure 4c).

iii) Bees find more flowers at intermediate densities

Finally, this result is based on a clear internal inconsistence in the work. As discussed in the manuscript, this optimal intermediate density appears because the homing mechanism is activated after a single flower is encountered, so producing the "masking effect". The authors justify this by saying that the bees will return to the nest to uncover their crop after each encounter. So, implicitly they are assuming that every single encounter is successful, it is, the crop is gathered after every single encounter. In consequence, it is inconsistent to say that "more successful foraging" is reached at intermediate flower densities just because more flowers are being encountered there. This is meaningless in a model where every encounter is assumed to be successful. To assess correctly this question it should be assumed that after a successful encounter the flower is exhausted, and so subsequent encounters with it (at least for a transient time) will be unsuccessful. Then an adequate measure of foraging efficiency could be conveniently proposed and evaluated.

In conclusion, while the present work seems to be technically and methodologically correct, I cannot recommend its publication due to a clear lack of novelty and relevance in the results/discussion reported.

Reviewer #3: This article studies how bees search for flowers by developing an individual-based realistic model and comparing with data. The model of bee search movement takes into account the returning to the nest, i.e., the tendency of these insecdts to make loops around their nest. One of the important results obained is the masking effect, so that detection of flowers close to others can be reduced. These are very nice results, novel and the authors try to connect with realistic measurments. Moreover, the paper is very clearly written and thus I recommend this work for publication after the authors have addressed the following minor corrections/suggestions:

1- The average value of tau, the switching time, is 1/preturn. In their calculations this is 30 seconds. Is this time enough for the exploratory phase when the bees move some kilometers (as it happens some times)? Please clarify.

2. Fig 2 is labelled with Fig.3. Please correct.

3. In the section of "Calibration with experimental data" the authors talk about "2D exploratory flight trajectories", but the data is for both exploratory and return phases. Am I correct? Please clarify.

4. The fit to optimal parameters with experiment should be clarified in the appendix. However, there are nowadays many techiniques coming from maching learning (gradient descent and variants) to minimize cost functions, and obtain optimal fittings. Not for this article, but the authors should consider (and comment) this possibility for future works.

5. Eq. M2 is not correct. It should be v(t+Delta t). Right?

6. The very interesting effect of masking reminds me that of "excess or defect of information is bad for searching of resources" as obtained in some articles of foraging (see for example, Martinez-Garcia et al, Physical Review Letters (2013)). Maybe the authors consider this paper woth of dicussing in their Discussion section.

7. They may also consider to discuss some extensions of their model to include the interaction of the bees. I find this very interesting for the future.

**Have the authors made all data and (if applicable) computational code underlying the findings in their manuscript fully available?**

Reviewer #1: Yes

Reviewer #2: None

Reviewer #3: Yes

PLOS authors have the option to publish the peer review history of their article (what does this mean?). If published, this will include your full peer review and any attached files.

Reviewer #1: No

Reviewer #2: No

Reviewer #3: No
---

## [Decision Letter · Decision Letter 1]

19 Jan 2023

Dear Mrs Morán:

Thank you very much for submitting your manuscript "Modeling bee movement shows how a perceptual masking effect can influence flower discovery" (PCOMPBIOL-D-22-01347R1) for review by PLOS Computational Biology. 

As with all papers reviewed by the journal, your manuscript was reviewed by members of the editorial board and by several independent reviewers. Although two of the Reviewers were positive about your work, Rev. #2 kept his initial criticism about the broader impact of your study. Based on these comments, we regret that we will not be pursuing this manuscript for publication at PLOS Computational Biology.

The reviews are attached below this email, and we hope you will find them helpful if you decide to revise the manuscript for submission elsewhere. 

While we cannot consider your manuscript further for publication in PLOS Computational Biology, we would like to offer you the option to transfer your submission, with reviews, to PLOS ONE https://www.editorialmanager.com/PONE/

If you DO wish to transfer your submission, please click this link:

<DeepLinkData><DeepLinkTypeID>27</DeepLinkTypeID><peopleID>1761025</peopleID><userSecurityID>aa560fb3-f18c-47bb-a843-91dac677af3f</userSecurityID><documentID>33314</documentID><revision>1</revision><manuscriptNumber>PCOMPBIOL-D-22-01347</manuscriptNumber><docSecurityID>1f41ade1-a9cc-47ca-9c5b-9351659fa5b3</docSecurityID></DeepLinkData>

If you do NOT wish to transfer your submission, please click this link to decline:

<DeepLinkData><DeepLinkTypeID>28</DeepLinkTypeID><peopleID>1761025</peopleID><userSecurityID>aa560fb3-f18c-47bb-a843-91dac677af3f</userSecurityID><documentID>33314</documentID><revision>1</revision><manuscriptNumber>PCOMPBIOL-D-22-01347</manuscriptNumber><docSecurityID>1f41ade1-a9cc-47ca-9c5b-9351659fa5b3</docSecurityID></DeepLinkData>

Please note, all PLOS journals are editorially independent and vary in submission requirements.

Should you choose to transfer, your manuscript files, along with the reviewers' comments and their identities will be transferred automatically, and you will receive a confirmation email within 24 hours. Once transferred, your submission will be returned to you so you can check over your record before completing the submission. You may be asked to provide additional information, such as a response to the reviewers' comments. If you have any questions, please contact the editorial office of PLOS ONE https://www.editorialmanager.com/PONE/

We are sorry that the news is not more positive on this occasion, and we hope you will consider PLOS Computational Biology for future submissions. Thank you for your support of PLOS and of open-access publishing.

Sincerely,

Ricardo Martinez-Garcia

Academic Editor

PLOS Computational Biology

Natalia Komarova

Section Editor

PLOS Computational Biology

Reviewer's Responses to Questions

**Comments to the Authors: **

Reviewer #2: After reading the second version of the manuscript I see that the authors have taken some time to address the criticisms I raised in my previous report. The main changes they have included are: (i) they have introduced a "Model background" section to incorporate the literature on random-walk models related to the central-place foraging problem, and (ii) they have included several figures (in particular Fig. 5B and Fig. S1) exploring additional aspects of the model. 

In spite of these changes, which have undoubtedly served to improve the manuscript, my general opinion remains the same. I do not see that the results presented in the work have a broad interest since they still remain mainly qualitative and particular. For example:

(i) it is not proved anywhere that the model really leads to stationary distributions (which is essentially the main novelty that the model is intended for). The arguments provided are apparently based just on visual inspection of the results, but I do not see in fig 3D that the distributions saturate, and actually in Fig. S1 the authors mention that for some particular choices of parameter values, non-saturation is observed. What is the real applicability of the model if its main property cannot be clearly stated? 

(ii) In any case, there are actually models (mentioned now in the 'Model background' section) that can be rigurously proved to saturate, and against which the present simulations could be tested. Do the decays observed in the simulations here agree with those predicted from those models? If so, ideas from those models (albeit being more simplistic) could be used in connection to the present one. Why do not trying to merge the present ideas to existing models (or compare them against) instead of presenting a model as if it has no relation at all with all the existing literature?

To summarize, behind all this there is the underlying question: what are models useful for? In my opinion, models are used, for example, in physics to reach conclusions as general as possible. In biology, on the contrary, they are used to understand or predict what are (or could be) the real mechanisms driving function or behavior. I do not see that the work and the results as they stand here fulfil any of these purposes, so that is why I cannot see their general interest.

Reviewer #3: The authors have properly addressed all my requirements and suggestions. Thus I consider this paper ready for publication.

**Have the authors made all data and (if applicable) computational code underlying the findings in their manuscript fully available?**

Reviewer #2: Yes

Reviewer #3: None

PLOS authors have the option to publish the peer review history of their article (what does this mean?). If published, this will include your full peer review and any attached files.

Reviewer #2: No

Reviewer #3: No

---

## [Editor Report · Decision Letter 2]

1 Mar 2023

Dear Mrs Morán,

We are pleased to inform you that your manuscript 'Modeling bee movement shows how a perceptual masking effect can influence flower discovery' has been provisionally accepted for publication in PLOS Computational Biology.

Best regards,

Ricardo Martinez-Garcia

Academic Editor

PLOS Computational Biology

Natalia Komarova

Section Editor

PLOS Computational Biology

---

## [Editor Report · Acceptance letter]

14 Mar 2023

PCOMPBIOL-D-22-01347R2 

Modeling bee movement shows how a perceptual masking effect can influence flower discovery

Dear Dr Morán,

I am pleased to inform you that your manuscript has been formally accepted for publication in PLOS Computational Biology. Your manuscript is now with our production department and you will be notified of the publication date in due course.

With kind regards,

Zsofi Zombor
